# Essential Oils and Antagonistic Microorganisms as Eco-Friendly Alternatives for Coffee Leaf Rust Control

**DOI:** 10.3390/plants12203519

**Published:** 2023-10-10

**Authors:** Maricela Santiago-Santiago, Gabriela Sánchez-Viveros, Luis Hernández-Adame, Cesar Josué Chiquito-Contreras, Alejandro Salinas-Castro, Roberto Gregorio Chiquito-Contreras, Luis Guillermo Hernández-Montiel

**Affiliations:** 1Posgrado en Ciencias Agropecuarias, Universidad Veracruzana, Xalapa 91090, Mexico; santiagomrc@gmail.com; 2Facultad de Ciencias Agrícolas, Universidad Veracruzana, Xalapa 91090, Mexico; gabsanchez@uv.mx (G.S.-V.); cchiquito@uv.mx (C.J.C.-C.); 3CONAHCYT-Centro de Investigaciones Biológicas del Noroeste, La Paz 23096, Mexico; ladame@cibnor.mx; 4Nanotechnology & Microbial Biocontrol Group, Centro de Investigaciones Biológicas del Noroeste, La Paz 23096, Mexico; 5Centro de Investigación en Micología Aplicada, Xalapa 91017, Mexico; asalinas@uv.mx

**Keywords:** action mechanism, biocontrol, *Hemileia vastatrix*, incidence, severity

## Abstract

Coffee leaf rust (CLR) is caused by the biotrophic fungus *Hemileia vastatrix* Berk. & Br., a disease of economic importance, reducing coffee yield up to 60%. Currently, CLR epidemics have negatively impacted food security. Therefore, the objective of the present research study is to show a current framework of this disease and its effects on diverse areas, as well as the biological systems used for its control, mode of action, and effectiveness. The use of essential plant oils and antagonistic microorganisms to *H. vastatrix* are highlighted. Terpenes, terpenoids, and aromatic compounds are the main constituents of these oils, which alter the cell wall and membrane composition and modify the basic cell functions. Beneficial microorganisms inhibit urediniospore germination and reduce disease incidence and severity. The antagonistic microorganisms and essential oils of some aromatic plants have great potential in agriculture. These biological systems may have more than one mechanism of action, which reduces the possibility of the emergence of resistant strains of *H. vastatrix*.

## 1. Introduction

Ethiopia is the center of origin of arabica coffee. Coffee has been cultivated in almost 85 countries, which covers a surface of more than 10 million hectares [1] and is the second most traded product at world level after petroleum and its derivatives [2,3]. Coffee is also the third most important food product after wheat (*Triticum* spp.) and sugar, generating more than 18,000 million USD/year and employing more than 125 million people around the world [4]. Despite the existence of more than 100 species of the genus *Coffea*, the coffee world production depends exclusively on two species: *Coffea arabica* L. and *C. canephora* [5], whose cultivation surface has decreased due to environmental problems, deforestation, soil degradation, pest incidence, and diseases, among others [4]. Among the coffee diseases, the most destructive for the crop is the coffee leaf rust (CLR), caused by the biotrophic fungus *Hemileia vastatrix* Berk. & Br. [6].

Coffee is the only host known for this fungus and is considered an obligate parasite because it cannot survive on soil or plant material. To date, its proliferation has not been possible in culture media in laboratory conditions [7,8]. Approximately 50 breeds of *H. vastatrix* have been reported [9], whose infection process starts when its urediniospores germinate and enter through the leaf stomata [10,11]. The disease causes physiological activity loss (photosynthesis, respiration, and transpiration), as well as aging and premature leaf and fruit loss before maturity [12]. Photosynthesis is the main physiological process affected by CLR [13]. The control of *H. vastatrix* is performed with the application of chemical products, such as synthetic fungicides based on copper [14].

Nevertheless, these chemical products generate environmental contamination and toxicity towards living beings, and increase crop production costs. Thus, the global need arises to propose more ecological and sustainable methods against CLR, such as biological control based on antagonistic microorganisms and the application of essential oils of several plants. These microorganisms have more than one mechanism of action: they decrease the appearance of resistant strains; they are not toxic to the environment nor to living beings, and they are biodegradable [15].

## 2. Dissemination of Coffee Leaf Rust (CLR) in the World

In 1869, CLR was reported for the first time in a plantation of Ceylon, now Sri Lanka [16]. In the same year, the British mycologist M.J. Berkeley and his assistant C.E. Broome described and named the causal agent of the disease as *Hemileia vastatrix* [17]. Its sudden form of appearance and rapid expansion in a very distant region from the coffee centers caused controversy on the disease origin [18].

The center of origin of a host and its phytopathogens usually coincide; thus, most likely the fungus was first detected in Africa [19]. However, CLR was introduced to Ceylon from Africa through infected plants, where the disease found favorable conditions for its development, causing outbreaks that were never recorded. From Ceylon, CLR dispersed rapidly in all Asia [20] and was simultaneously detected in all the eastern African countries, but it took time to reach western Africa. This tardy arrival may explain why CLR was unknown in America until 1970 [21]. 

The theory of its dissemination is that *H. vastatrix* urediniospores were transported by the trade winds from western Africa to Brazil. Moreover, a theory mentions that the disease was introduced accidentally through plant or material or the contaminated clothes of the coffee grain producers or collectors [22]. In a period of 20 years after its arrival in Brazil, CLR was detected in all the coffee producers in Latin American countries [23]. In 2012, all the countries in Central America, the Caribbean, and Mexico were strongly affected by the epidemics, which caused losses of more than 500 million USD. This situation affected more than 300 thousand producers, and in 2013, the effect was on Peru and Ecuador [17]. In Brazil, CLR was identified for the first time by the researcher Arnaldo Gomes Medeiros in southern Bahia in January 1970 [24].

### 2.1. Economical Affectation of Coffee

CLR has caused economic harvest losses throughout time. Before its detection in Ceylon, the first coffee producer in the world, its exports reached 41,855 tons (t) in 1860 but went down to 9000 t in 1879. By 1884, only 2300 t were exported [25]. During the CLR epidemics in Colombia (2008–2011), its production went down to 31% compared to that produced the year before. In the case of Central America, production went down to 10 and 16% during the years of epidemics [17].

These decreases directly impacted the coffee economy, affecting thousands of small producers and grain collectors. In the zones of Central America, coffee is the only source of income used for buying food and agricultural supplies; unfortunately, the epidemics have had an impact on food security [26].

### 2.2. Disease Cycle

CLR is recognized by the appearance of yellow or orange spots in leaves (Figure 1a), generating more than 70% of leaf loss (Figure 1b) and reducing plant yield to 60% [9,10]. The first macroscopic symptoms in the plants are chlorotic spots [27] and round lesions 1–3 mm in diameter, which grow when sporulation starts (spore formation on the leaf underside). When the CLR spots age, the orange powder turns to pale orange. After that, a brown or black spot appears on the center of the lesion with a dry aspect, which grows until it covers the entire lesion surface [28].

The main *H. vastatrix* multiplication form is the urediniospores, but it also produces teliospores and basidiospores [21,29]. This fungus shows an exclusive dependence on the plant. Since no other host exists for this phytopathogen, *H. vastatrix* feeds on live leaf cells to grow and reproduce itself [30]. The CLR disease cycle is very simple (Figure 2).

The first stage starts with urediniospore dissemination that occurs with the liberation, dispersion, and inoculation of fungal spores. The second stage is the germination and incubation periods [31]. When the leaf surface remains humid for several hours, the spores can survive longer or germinate and penetrate the leaves [32]. The third stage starts with the fungus penetration through the leaf stomas, followed by leaf colonization through the hyphae. Once the hypha or appressorium gets inside the leaf, fungi develop structures called haustoria that contact plant cells and extract the nutrients for their growth [33]. The fourth stage is the sporulation period, in which the sporophore emerges and new infectious urediniospores are produced. The period from germination to sporulation is called the latent period, which is of utmost importance in the entire CLR life cycle since the shorter it is, the greater possibility the cycle repeats itself and thus the more severe the result [34]. The CLR polycycle has from six to eight life cycles in one year, and depending on the environmental conditions and region, its reproduction is sexual, asexual, or crypto-sexual (hidden sexuality) [35].

Coffee leaf rust epidemics show three phases; the slow phase shows infection of few leaves; the second one is fast or explosive; and the third one is terminal or maxima. In the first phase, the primary inoculum is formed, and subsequently, the disease pathological cycle is repeated [36].

## 3. Coffee Leaf Rust Management

The control of coffee leaf rust (CLR) is performed with the use of agrochemicals and synthetic fungicides based on copper, biofungicides, resistant varieties, biological control, cultural control, and integrated management [10]. According to the incidence and severity of the disease, different control measures are established [37]. When the presence of fungal urediniospores—known as yellow or orange dust on the underside of the leaf—is visualized, eradication and isolation measures are applied, which consist of establishing security strips of approximately 50 km in width to separate the infected areas from the main producer regions. All the infected plants are cut and burned [24]. Chemical control is the commonly used strategy for CLR, applying chemical products as copper oxychloride and mineral broths [38]. Synthetic fungicides are also used, such as cyproconazole, flutriafol, thiamethoxam, triadimenol, imidacloprid, and propiconazole, among others [39], but they have lost effectiveness due to the resistance developed by the phytopathogen [40]. Moreover, climate change with variation in temperature and rain patterns has been a key factor in *H. vastatrix* development and strengthening [9,41]. Another factor that adds to the disease propagation is the predominance of coffee varieties with few resistant genes. The factors that increase harm caused by *H. vastatrix* are inadequate shade management, abundant weeds, old plantations, and plants with nutritional deficiencies [42].

The timing of synthetic fungicide application is critical for controlling CLR. The maximum effect has been achieved through applications before it starts and during the early period of the rainy season. These chemicals are either protectant or kill the fungus on coming into contact with it on the plant surface. Cultural practices can have an indirect but beneficial effect on CLR control. Wider spacing and appropriate pruning help prevent prolonged wetness and increase the penetration of applied fungicides. Proper nutrition management increases vigor and significantly reduces the CLR effects [8,9]. 

## 4. Eco-Friendly Alternatives in the Sustainable Management of Coffee Leaf Rust (CLR)

### 4.1. Resistant Plants

Currently, the use of varieties genetically resistant to CLR is considered the most efficient strategy for its control. The species *C. canephora* provides the greatest source of genetic resistance to *H. vastatrix* [43], which has been transferred to commercial varieties of *C. arabica* by crossbreeding the Timor hybrid. Disease resistance is a condition of the defense system recognizing 1 of the 50 *H. vastatrix* breeds among the interaction between the nine genes of the specific resistance R (SH, 1–9) in the host with the virulence factors (V, 1–9) of the phytopathogen [6]. Nevertheless, coffee quality is sometimes inferior to that of the traditional varieties [44]. To a lesser extent, biological control is used as treatment against the disease. A set of secondary plant microorganisms and metabolites show antifungal activity against this phytopathogen [45].

### 4.2. Secondary Plant Metabolite for the Control of Coffee Leaf Rust (CLR)

Different studies have reported the efficiency of plant extracts for controlling CLR [2,3,10,12,14,44,45], which have low molecular weight compounds [46] and products of different metabolic plant routes [47]. These products can be distinguished from the primary metabolites to the amino acids, fats, lipids, and carbohydrates, which are not essential to maintain life, but they are for the survival of an organism; they are extremely diverse and are identified in several classes. The extracts have structure functions, fuel, signaling, stimulating, and inhibiting effects on enzymes, defense, and interaction with other organisms [48,49]. The secondary plant metabolites have an outstanding function in protection against microbial predators and phytopathogens due to their toxic nature and repellence to herbivores [50]. Different techniques can be applied when using plant species to obtain compounds, also called bioactive (because of the compound function when used in pure form). Some of them are extraction technology with supercritical fluids, organic solvents, or assisted by microwaves, and extraction equipment with assisted ultrasound or with a distillation device. The products of these extractions and distillations are generally extracts and essential oils that have antifungal properties (Table 1), such as insecticides, antibacterials, antiviral and resistance inductors [51]. Plant extracts, such as *Baccharis glutinosa*, *Camellia sinensis*, *Eriobotrya japonica*, *Ardisia compressa*, and *Ocimun basilicum* have a significant effect on CLR spore germination inhibition and disease severity reduction [52].

#### 4.2.1. Main Plants as Sources of Bioactive Compounds

The extraction of bioactive compounds starting from natural sources is a method much used in industries, mainly including in pharmaceutics, cosmetics, and food. Among these natural sources, aromatic plants—also known as herbs and species—are the most studied. Since ancient times, these plants have been used in traditional medicine and as additives in food [57]. The secondary metabolites can be divided into three groups according to their biosynthetic origin: terpenoids, nitrogen compounds, and phenylpropanoids, also known as phenolic compounds [58]. Plant species such as *Cymbopogon nardus* (citronella grass), *Cymbopogon citratus* (lemmon grass), *Corymbia citriodora* (eucalyptus, lemon-scented gum), *Melaleuca alternifolia* (tea tree), *Thymus vulgaris* (thyme), *Azadirachta indica* (neem), *Syzygium aromaticum* (clove), and *Allium sativum* (garlic) have been studied to obtain phytoextracts that show antifungal properties against *H. vastatrix* [54]. Other plant species have also been studied to obtain bioactive compounds, such as *Cinnamomum verum* or *C. zeylanicum* (cinnamon), *Citrus sinensis* (orange), *Larrea tridentata* (creosote bush), *Eucalyptus globulus* (Tasmaninan bluegum), *Brassica nigra* (black mustard), *Piper nigrum* (black pepper) [52], *Tribulus terrestris* (puncture vine), *Datura ferox* (fierce thornapple), *Mansoa alliacea* (wild garlic), *Ricinus communi* (castor bean), *Acacia farnesiana* (acasia), *Cestrum* sp. (lady of the night), *Zingiber officinale* (ginger) [59], *Piper aduncum* L. (spiked pepper) [60], *Garcinia gardneriana* (Bacupari, tree native to Brazil) [61], and *Baccharis salicina* (willow baccharis) [62]. Three of the botanical extracts evaluated were aloe vera or leaf juice (*Aloe barbadensis*), moringa (*Moringa oleífera*), and tobacco (*Nicotiana tabacum*) [63].

#### 4.2.2. Chemical Composition

Aromatic plants contain chemical compounds, such as polyphenols, quinines, flavonoids, alkaloids, polypeptides, and terpenoids, among others [57]. These types of plants have higher concentrations of bioactive compounds [50], also known as phytochemicals with biological activity [64]. The phenolic compounds, also known as polyphenols or phenylpropanoids are the most abundant secondary plant metabolites. The support structure or chemical scaffold of these compounds is in the aromatic ring with one or more hydroxyl groups. Their classification is based on the number of phenol molecule units [50]. The terpenes are aromatic organic compounds, unsaturated hydrocarbons that have a chemical and structural arrangement different from the simple isoprene molecule. Terpenes may be classified as monoterpenes, sesquiterpenes, diterpene, triterpenes, tetraterpenes, sterols, or terpenoids [65]. The nitrogen compounds, also known as alkaloids, are biomolecules constituted by nitrogen and found only in some essential oils. Because of their characteristic aroma, they are used as a floral fragrance fixative and as flavoring agents in ice cream and cigarettes [66]. Since essential oils are very complex mixtures, they may contain more than 20 components in different concentrations. Terpenes, terpenoids, and aromatic compounds are the main constituents, which make up from 20 to 70% of the total concentration and the rest comprises the major components [67]. Table 2 shows the most representative chemical composition of the bioactive compounds.

#### 4.2.3. Mechanisms of Action

The plants produce biomolecules during their secondary metabolism, such as terpenes, phenols, quinones, polyacetylenpos, polyenes, alkylamines, carbohydrates, organic acids, alkaloids, lectins, or peptides that alter the cell wall and membrane composition and modify the basic cell functions [59]. The reported in vitro studies mainly characterize essential oils as antioxidant compounds. However, they have also been reported to act as prooxidants in eukaryotic cells, affecting the internal cellular membranes and organelles such as mitochondria, while the plant extracts inhibit fungal radial growth and spore germination and formation [53]. Depending on the concentration and source, they show cytotoxic effects on live cells but not genotoxic effects. This activity is due mainly to the presence of alcohols, phenols, and aldehydes [75]. According to some authors, the fungal mode of action of the oils consists of inhibiting the quinine component synthesis of the fungal cell wall [76].

Some essential oils have a phytotoxic effect, and their potential use is recommended in formulating bioherbicides, as well as being used as elicitors. Phytotoxicity is an inconvenience for its potential applicability. Thus, some authors have suggested studying the identification of their mechanism of cellular action, mainly identifying the molecular target [77,78]. However, in the case of crops, experiments performed in greenhouses have not shown any phytotoxicity symptoms caused by the application of essential oils in plants [79]. Figure 3 shows the proposal of applying both biological systems (microorganisms and essential oils) in coffee plants, as well as the plant–pathogen interaction.

The bacterial mechanism of action has been studied in both essential oils (oregano and clove) and their major phenolic components (thymol and eugenol) on *Bacillus subtilis*, finding that both oils and phenolic compounds were able to induce cellular lysis [80]. The essential oils can also have an elicitor effect, that is, induce a signal to activate the mechanisms of plant defense [81].

## 5. Antagonistic Microorganisms to *Hemileia vastatrix*

Beneficial microorganisms constitute an alternative to promote plant growth and productivity, improving their resistance and protection towards phytopathogens; moreover, these microorganisms do not contaminate the environment and they are easy to apply with a reduced cost [82,83]. Biocontrol has been proposed in organic coffee cultivations as an alternative to CLR control. The bacteria of the genus *Pseudomonas* have generated a great interest in this field because of their benefits for the host plant by producing auxins and siderophores that improve plant development and growth as well as protecting them against phytopathogens [84]. Studies have discovered that *Bacillus subtilis* and *Pseudomonas fluorescens* inhibit urediniospore germination and reduce disease incidence and severity. Other microorganisms act as myco or hyperparasites, such as the fungus *Lecanicillium* spp. [85].

The biocontrol method has been used around the world for more than a century [86]. Mainly natural and recombinant strains of the genera *Bacillus*, *Pseudomonas*, and *Trichoderma* are commercially available as biocontrol agents. Furthermore, the secondary metabolites—derivatives of the complete microorganisms—are also used [87,88]. *Trichoderma* spp. are one of the most successful biocontrol agents in agriculture because of their efficiency and production facility. They act as biopesticides, biofertilizers, growth promoters, and natural resistance inductors [89].

### 5.1. Bacteria

In coffee production, different microorganisms are used mainly as plant growth promoters. The most used genera are: *Alcaligenes*, *Pseudomonas*, *Azospirillum*, *Bacillus*, *Klebsiella*, *Azotobacter*, *Enterobacter*, *Gluconacetobacter*, *Burkholderia*, *Arthrobacter*, *Rhizobium*, *Bradyrhizobioum*, and *Serratia*, which may improve nutrient absorption, root and shoot formation, environmental stress tolerance, phosphorus solubilization and fixation, and phytohormone secretion, and control plant phytopathogens [90]. Bacterial strains, such as *B. thuringiensis* [91], *B. lentimorbus* [62], *B. cereus* [92], *Brevibacillus choshinensis* [52], *Salmonella enterica* [93], *P. fluorescens,* and *B. subtilis* [85] have been reported as antagonists to the causal CLR agent.

### 5.2. Fungi

*Trichoderma* spp. are the main fungus genus used in biocontrol and are considered the most studied as antagonistic to CLR [94]. Some authors have reported antifungal activity by inhibiting *H. vastatrix* spore germination through the action of chitosan oligomers of some fungal classes of *Basidiomycetes*, *Ascomycetes*, *Zygomycetes,* and *Deuteromycetes* [52]. Moreover, fungi as *Fusarium* spp. are phytopathogens, but they have been reported as antagonists to the CLR phytopathogen (Table 3).

### 5.3. Mechanisms of Antagonic Action

Biocontrol with microorganisms may be a viable and environmentally safe alternative compared with synthetic fungicides. These microorganisms use one or several antagonistic mechanisms to control phytopathogens such as CLR and are classified as direct and indirect. The main antagonistic mechanisms of fungi and bacteria are competition for space and nutrients, inhibition for organic volatile compounds, organics, siderophore production, antibiotics, hydrolytic enzymes, and induction to host resistance. The direct mechanisms are those where the microorganism synthesizes antibiotics or other compounds that exert an inhibiting effect against the phytopathogen [95]. 

The bacteria of the genus *Bacillus* and *Pseudomonas* mainly synthesize a great number of hormones that influence plant growth and development, as in the protection against phytopathogens. Generally, these species are used as biocontrol agents of plant diseases. The most frequent mechanisms associated with the antagonistic effect of these species are the production of compounds with antimicrobial activity and the induced systemic response (ISR). The *Bacillu*s strains produce a great variety of antifungal secondary metabolites, mainly lypopeptides, such as surfactine, fengicine, and iturine that act mainly on the cell wall. On the other hand, *Pseudomonas* species produce antibiotics such as 2,4–diacethylfloroglucinol [96,97,98]. 

Other biocontrol strategies against phytopathogens are the use of microparasites, fungi that have the capacity to survive at the expense of another fungus (Table 3). They also act as hyperparasites, where the mycelium of the biological control penetrates in different structures of the phytopathogen and partially degrades its cells by the action of enzymes such as chitinases, glucanases, and proteases [52]. 

**Table 3 plants-12-03519-t003:** Antagonistic microorganisms to *Hemileia vastatrix* in coffee.

Microorganism	Control	References
*Trichoderma* spp.	Hyperparasitism Lytic enzyme production (chitinases and glucanases) Induced systemic resistance (ISR)	[94]
*Pseudomonas putida* P286 *Bacillus thuringiensis* B157	Secondary metabolite production (antibiotics, antifungal and volatile organic compounds) Induced systemic resistance (ISR)	[96]
*Bacillus* spp. (B10, B25, B143, B157, B171, B175) *Pseudomonas* sp. (B286)	Toxic compound production that inhibits *Hemileia vastatrix* urediospores germination	[97]
*Penicillium* sp. *Aspergillus* sp.	Mycoparasitism	[97]
*Lecanicillium* sp. *Calcarisporium arbuscula* *Sporothrix* sp. *Simplicillium* sp.	*In vitro* mycoparasitism in CLR urediospores	[99]
*Verticillium hemileiae* Bouriquet	Hyperparasitism in greenhouse and field plants	[100]
*Fusarium* spp. *Acremonium byssoides* *Verticillum lecanii* *Chrysosporum ovalisporum* *Fusarium pallidoroserum*	Mycoparasitism Lytic enzyme production (chitinases and glucanases)	[101]

One of the most known mechanisms of beneficial bacteria against *H. vastatrix* is competition for space or nutrients; many types of bacteria live both within (endophytes) and on (epiphytes) plant tissues and their presence avoids phytopathogen germination and development. A second mechanism is antimicrobial metabolite production, such as hydrolytic enzymes, which degrade the fungus cell wall. The third one is plant systemic resistance induction [52,102]. In the case of *Trichoderma* spp., also known as colonizers established in plant roots, they are very powerful biocontrol agents that induce ISR in plants. Their main mechanisms are mycoparasitism and antibiosis against several phytopathogens [103].

## 6. Cost-Benefit of Applying Essential Oils and Microorganisms for Controlling Coffee Leaf Rust (CLR)

Essential oils, also known as volatile oils, are substances produced by the plant’s secondary metabolism. Because of their bactericidal, fungicidal, virucidal, insecticidal, acaricidal, nematicide, and herbicidal properties, these compounds have been studied more than ever to be included as bio-pesticides in the agricultural sector [77]. The essential oils are extracted from aromatic plants using different methods. The main form of extraction is aromatic plant distillation [75]. These oils are mainly composed of volatile molecules, such as terpenes, terpenoids, phenol derivatives, and aliphatic components. For example, *Eucalyptus* sp. essential oils have been widely used as bioactive agents in different crops of interest. Some studies have reported the evaluation of in vitro and in vivo antagonistic activity of these oils against the CLR [54,56].

Moreover, a biofungicide is a product that contains live microorganisms showing beneficial properties for plant health and growth by acting against phytopathogens [104,105]. Recently, commercial biofungicides formulated with microorganisms and plant extracts such as Roya Out^®^ (50% *B. subtilis*, 2.5% *Azadirachta indica*, and 2.5% *Syzygium aromaticum*), Biogeneser^®^ (*Azadirachta indica*), and Timorex Gold^®^ (23.8% tea tree oil *Melaleuca alternifolia*) have been evaluated in field conditions for the control of CLR on sites where the disease has been observed year by year [54,106]. 

Pest control and diseases have two ways of implementation, by conservation or introduction. One option is by preserving the available biological control agents in nature to obtain ecosystemic benefits, whereas for the introduction method, natural enemies are bred and multiplied in the laboratory or bio-factories [107]. Thus, changing the chemical control methods used in agriculture to sustainable methods controlling phytopathogens should allow maintaining agroecosystems and ecological biodiversity. Although the application of chemical synthesis products offers immediate protection, it causes phytopathogen resistance and may bioaccumulate in crops [108]. The cost of using agrochemicals may be reduced up to 60% compared to the use of biological control agents [107].

## 7. Future Applications and Challenges

Just as disease appearance may alter the metabolism of coffee plants, causing changes in coffee grain composition and quality of the beverage [109], the different inputs used for disease control may also modify the organoleptic and fruit quality characteristics [1]. In the last years, research on the potential use of essential oils in agriculture has increased considerably. Because of their antimicrobial, fungicidal, and bactericidal properties, they are promising candidates for developing new potentially eco-friendly products for an integral pest and disease control. Likewise, their phytotoxic properties may potentiate their application as herbicides in agriculture [110]. Nevertheless, most of the biocontrol applications and studies are limited to greenhouse crops where the environmental conditions are monitored and supervised. In field conditions, extensive studies have also been dedicated to evaluating the efficiency of the biological control agents. Nevertheless, this area has been restricted to changes in the ecological, genetic, and physiological status of the host and the climatological characteristics [108]. In conclusion, alternatives should propose the union of various strategies that protect biocontrol agents from extreme conditions, and their viability and bioavailability.

## 8. Conclusions

The antagonistic microorganisms and essential oils of some aromatic plants have great potential in agriculture. Efforts should be made to increase and generate scientific evidence in the application of biological systems integrated with antagonistic essential oils and microorganisms in commercial coffee plantations. Therefore, further studies are needed to identify and explain the different cellular and molecular mechanisms of action observed, mainly in essential oils on coffee plants to identify the molecular target due to the difficulty of achieving long-term treatment resistance for *H. vastatrix* variability. 

## Figures and Tables

**Figure 1 plants-12-03519-f001:**
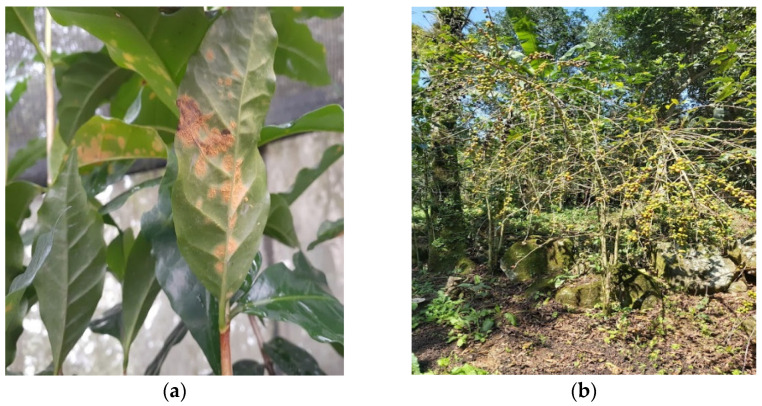
Symptoms and signs of the coffee leaf rust (CLR): (**a**) Chlorotic spots and urediniospores on the underside of the leaves; (**b**) Severe defoliation in a coffee plant because of *Hemileia vastatrix*.

**Figure 2 plants-12-03519-f002:**
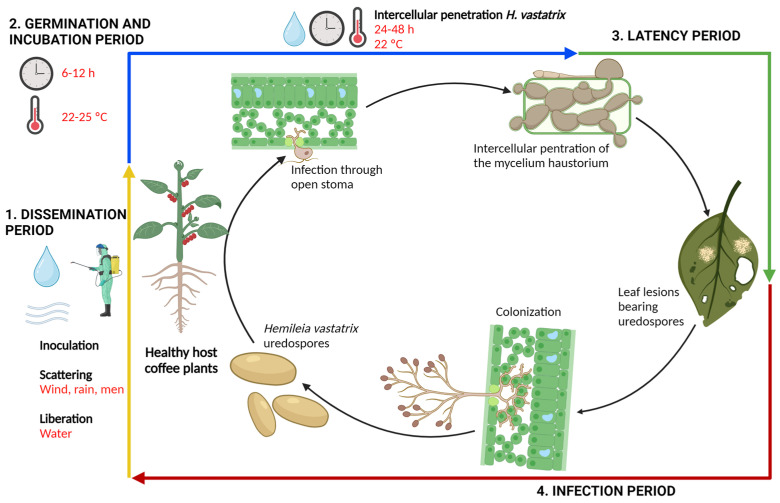
Life cycle of *Hemileia vastatrix*. The asexual cycle of coffee leaf rust (CLR) starts with urediniospore deposition on the leaves.

**Figure 3 plants-12-03519-f003:**
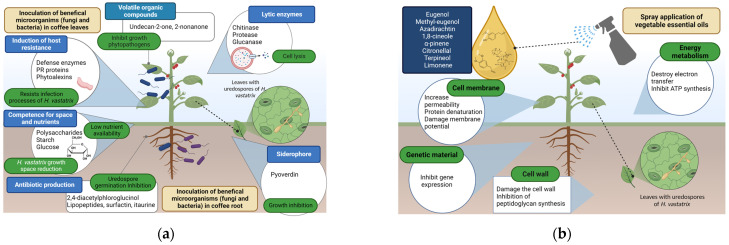
Application of biological agents to control *Hemileia vastatrix* in coffee plant: (**a**) Beneficial microorganisms; (**b**) Essential oils.

**Table 1 plants-12-03519-t001:** Essential oils with a fungicide effect against *Hemileia vastatrix*.

Essential Oil	Components	Control	References
*Ciannamomum verum* (cinnamon) *Citrus sinensis* (orange) *Brassica nigra–Piper nigrum* (black mustard–black pepper)	Trans-cinnamaldehyde, eugenol and cumarina Limonene, geraniol, nerol, geranyl acetate, caryophyllene, neryl acetate, inalol and cineol Allyl isothiocyanate Piperine, δ-3-carene, β-pinene, limonene, sabinene, β-caryophyllene, α-selineno and β-selineno and germecrene	Decrease in CLR severity in coffee plants var. Caturra and Typica	[53]
*Azadirachta indica* (neem)	Azadirachtin	Inhibition in vitro of *Hemileia vastatrix* urediniospores germination	[54,55]
Eucalyptus (*Eucalyptus citriodora*, *Eucalyptus grandis*, and *Eucalyptus microcorys*)	1,8-cineole, α-pinene, citronellal, terpineol, limonene	Antifungal in vitro and antagonist in vivo activities against *H. vastatrix*	[56]
*Melaleuca alternifolia* (tea tree) *Cinnamomum zeylanicum* (cinnamon) *Cymbopogon citratus* (lemongrass) *Cymbopogon nardus* L (citronella) *Syzygium aromaticum* (clove) *Corymbia citriodora* (lemon eucalyptus) *Thymus vulgaris* (tyme)	Secondary metabolites (Alkaloids, flavonoids, terpenoids, and tannins)	Inhibition in vitro of urediniospores germination of *H. vastatrix* Partial control CLR in coffee plants var. Mundo Novo and Catuaí Promoting cellular disorganization and vacuolization in urediniospores of *H. vastatrix*	[10,55]

Abbreviation: CLR = coffee leaf rust.

**Table 2 plants-12-03519-t002:** Classification and chemical composition of the main secondary plant metabolites.

Group	Chemical Component	Natural Source	Location	References
Terpenes	Eucalyptol (1,8-Cineole) 1,3,3-trimethyl- 2-oxabicyclo [2,2,2] octane C_10_H_18_O	Eucalyptus, bay leaves, marihuana, tea tree	Essential oil	[68]
	Limonene 1-methyl-4-(1-methylethenyl)-cyclohexene C_10_H_16_	Eucalyptus, citrics	Essential oil	[69]
	Carvacrol o cimofenol 5-isopropyl-2-metylphenol C_10_H_14_O	Oregano, thyme	Essential oil	[70]
Phenolic compounds (phenylpropanoids or polyphenols)	Eugenol 4-Allyl-2-methoxyphenol C_10_H_12_O_2_	Clove, cinnamon, turmeric, oregano, nutmeg	Essential oil	[71]
	Cinnamaldehyde 3-phenyl-2-propenal C_9_H_8_O	Cinnamon	Essential oil	[72]
Nitrogen or alkaloid compounds	Indole 2,3-Benzopyrrol C_8_H_7_N	Neroli and citric fruits	Extracts and essential oils	[73]
	Nicotine (S)-3-(1-methylpyrrolidin-2-yl) pyridine C_10_H_14_N_2_	Tobacco	Extracts and oils	[74]

## Data Availability

Not applicable.

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
