# Peer review of "Essential Oils and Antagonistic Microorganisms as Eco-Friendly Alternatives for Coffee Leaf Rust Control"

_plants, 2023, doi:10.3390/plants12203519_

Round 1

Reviewer 1 Report

This review evaluates the different soulutions on biocontrol with microorganisms or essential oil to control coffee leaf rut.

The review is clearly structured with and overview of the solutions. The references are pertinent.

So minor corrections can be found in the attached document.

I recomend to extend a little more the abstract showing the solutions available to fiht the leaf rut and some conclusions. Now it ooks more like a little introduction

The are few minor errors

Author Response

REVIEWER: 

“So minor corrections can be found in the attached document.” “I recomend to extend a little more the abstract showing the solutions available to fiht the leaf rut and some conclusions. Now it ooks more like a little introduction.” “Please complete your abstract with some of your results and conclusion.”    

ANSWER:

The abstract (Lines 17-27) changed: “Abstract: The coffee leaf rust (CLR) is caused by the biotrophic fungus Hemileia vastatrix Berk. & Br. -a disease of economic importance- reducing coffee yield up to 60%. Currently, CLR epidemics has negatively impacted food security. Therefore, the objective of the present research study is to show a current framework of this disease and its effects on diverse areas, as well as the biological systems used for its control, mode of action and effectiveness. The use of essential plant oils and antagonistic microorganisms to H. vastatrix are highlighted” by “Abstract: The coffee leaf rust (CLR) is caused by the biotrophic fungus Hemileia vastatrix Berk. & Br. a disease of economic importance- reducing coffee yield up to 60%. Currently, CLR epidemics has negatively impacted food security. Therefore, the objective of the present research study is to show a current framework of this disease and its effects on diverse areas, as well as the biological systems used for its control, mode of action and effectiveness. The use of essential plant oils and antagonistic microorganisms to H. vastatrix are highlighted. Terpenes, terpenoids and aromatic compounds are the main constituents of these oils, which alter the cell wall and membrane composition and modify the basic cell functions. Beneficial microorganisms inhibit urediniospore germination and reduce disease incidence and severity. The antagonistic microorganisms and essential oils of some aromatic plants have a great potential in agriculture. These biological systems may have more than one mechanism of action, which reduces the possibility of the emergence of resistant strains of H. vastatrix.”       

Reviewer 2 Report

The authors summerizes improtant role of Essential Oils and Antagonistic Microorganisms Alternatives for Control of Coffee Leaf Rut. The manuscript provides good information about the role and mechansim of essential oil and microorganism. This MS may be acepted to publish.  

Authors should check the english speeling and typo errors throughout the MS.

Author Response

REVIEWER :

"Authors should check the English speeling and typo errors throughout the MS.”

ANSWER:

Was checked the English speeling and typo errors in the manuscript.

The changes were:

  1. The word “of” is removed from the title: “Essential Oils and Antagonistic Microorganisms as Eco-friendly Alternatives for Coffee Leaf Rut Control”.
  2. Line 37, changed “exist” by “the existence of”.
  3. Line 57, changed “which” by “they”.
  4. Line 61, the word was added “the”.
  5. Line 78-79, changed “affecting” by “affected” and the word was added “This situation”.
  6. Line 80, changed “it” by “CLR”.
  7. Line 103, changed “for” by “of”.
  8. Line 104, changed “is” by “are”.
  9. Line 108, changed “urediniospores” by “urediniospore”.
  10. Line 112, changed “of the leaf” by “leaf”.
  11. Line 124, the word was added “the”.
  12. Line 136, the word was added “measures”.
  13. Line 137, changed “kilometers” by “km”.
  14. Line 141, the word was removed “such”.
  15. Line 148, changed “fungicides” by “fungicide”.
  16. Line 160, changed “being” by “has been”.
  17. Line 156, the word was added “by”.
  18. Line 169, changed “of CLR” by “CLR”.
  19. Line 167, the word was delated “that”.
  20. Line 169, the word was delated “of”.

Reviewer 3 Report

The authors revise an interesting topic about the biological control of Coffee Leaf Rut. However, there are some details that should be revised. 

In a first instance the Ms is centered about the disease of coffee produced by the fungus Hemileia vastatrix, being very interesting since coffee is a very important crop worldwide. However, the section 5 Antagonistic Microorganisms to Hemileia vastatrix is written very general, about the biological control strategies and the most used microorganisms in biological control strategy but little is mentioned about specifically control of  Coffee Leaf Rut. In this section 5.1 has no sense since there is not 5.2 and in general this complete section 5 should be reestructured.

On the other hand, sections 3, 6 and 7 are very short therefore they should be arranged or reestructured.

The refrences should be checked since several of them are not related with the content of the sentence, for example in line 39 reference 7. In this reference the authors did not study the culture media, even they did not mentioned anything in the article about the impossibility to growth the fungus in laboratory conditions, therefore the authors must put a valid reference for this important point or delete that this fungus has not been possible proliferate in laboratory conditions, since it is very important point for the researchers who want to growth this fungus.

Line 296 More recent Reference should be added: https://doi.org/10.1007/s00248-022-02044-2.

Line 309 this reference is about bacillus lipopeptides not related with ISR

The english should be checked since there are grammar mistakes such as lines 26, 37, 49, 54, 107 among others.

The english should be checked since there are grammar mistakes such as lines 26, 37, 49, 54, 107 among others.

Author Response

Reviewer 3:

  1. “The section 5 Antagonistic Microorganisms to Hemileia vastatrix is written very general, about the biological control strategies and the most used microorganisms in biological control strategy but little is mentioned about specifically control of Coffee Leaf Rut. In this section 5.1 has no sense since there is not 5.2 and in general this complete section 5 should be reestructured.”

ANSWER: Section 5 was restructured (Lines 263-281) “5. Antagonistic Microorganisms to Hemileia vastatrix

Beneficial microorganisms constitute an alternative to promote plant growth and productivity, improving their resistance and protection towards phytopathogens; moreo-ver, these microorganisms do not contaminate the environment, they are easy to apply with a reduced cost [82,83]. Biocontrol has been proposed in organic coffee cultivations as an alternative to CLR control. The bacteria of the genus Pseudomonas have generated a great interest in this field because of their benefits on the host plant by producing auxins and siderophores that improve plant development and growth besides protecting them against phytopathogens [84]. Studies have discovered that Bacillus subtilis and Pseudomo-nas fluorescens inhibit urediniospores germination and reduce disease incidence and sever-ity. Other microorganisms act as myco or hyperparasites as the fungus Lecanicillium spp. [85].

The biocontrol method has been used around the world for more than one century [86]. Mainly natural and recombinant strains of the genera Bacillus, Pseudomonas, and Trichoderma are commercially available as biocontrol agents. Furthermore, the secondary metabolites -derivatives of the complete microorganisms- are also used [87,88]. Trichoderma sp. are one of the most successful biocontrol agents in agriculture because of their efficien-cy and production facility. They act as biopesticides, biofertilizers, growth promoters, and natural resistance inductors [89].”

  1. “Sections 3, 6 and 7 are very short therefore they should be arranged or reestructured.”

ANSWER: All Sections were restructured.

Section 3 (Lines 130-154) “3. Coffee Leaf Rust Management

The control of coffee leaf rust (CLR) is performed with the use of agrochemicals and synthetic fungicides based on copper, biofungicides, resistant varieties, biological control, cultural control and integrated management [10]. According to the incidence and severity of the disease, different control measures are established [37]. When the presence of fungal urediniospores -known as yellow or orange dust on the underside of the leaf- is visual-ized, eradication and isolation measures are applied, which consist of establishing secu-rity strips of approximately 50 km in width to separate the infected areas from the main producer regions. All the infected plants are cut and burnt [24]. The chemical control is the commonly used strategy for CLR, applying chemical products as copper oxychloride and mineral broths [38]. Synthetic fungicides are also used, such as cyproconazole, flutriafol, thiamethoxam, triadimenol, imidacloprid, propiconazole, among others [39], but they have lost effectiveness due to the resistance developed by the phytopathogen [40]. Moreover, climate change with variation in temperature and rain patterns has been a key factor in H. vastatrix development and strengthening [9, 41]. Another factor that adds to the disease propagation is the predominance of coffee varieties with few resistant genes. The factors that increase harm caused by H. vastatrix are inadequate shade management, abundant weeds, old plantations, and plants with nutritional deficiencies [42].

The timing of synthetic fungicide application is critical for controlling CLR. The maximum effect has been achieved through applications before it starts and during the early period of the rainy season. These chemicals are either protectant, kill the fungus on coming into contact with it on the plant surface. The cultural practices can have an indi-rect but beneficial effect on CLR control. Wider spacing and appropriate pruning help preventing prolonged wetness and increasing penetration of applied fungicides. Proper nutrition management increase vigor and significantly reduces the CLR effects [8,9].”

Section 6 (332-359) “6. Cost-Benefit of Applying Essential Oils and Microorganisms for Controlling Coffee Leaf Rust (CLR)

Essential oils, also known as volatile oils, are substances produced by the plant sec-ondary metabolism. Because of their bactericidal, fungicidal, virucidal, insecticidal, acari-cidal, nematicide, and herbicidal properties, these compounds have been studied more than ever to be included as bio-pesticides in the agricultural sector [77]. The essential oils are extracted from aromatic plants using different methods. The main form of extraction is aromatic plant distillation [75]. These oils are mainly composed by volatile molecules, such as terpenes, terpenoids, phenol derivatives, and aliphatic components. For example, Eucalyptus sp. essential oils have been widely used as bioactive agents in different crops of interest. Some studies have reported the evaluation of in vitro and in vivo antagonistic ac-tivity of these oils against the CLR [54,56].

Moreover, a biofungicide is a product that contains live microorganisms showing beneficial properties for plant health and growth by acting against phytopathogens [104,105]. Recently, commercial biofungicides formulated with microorganisms and plant extracts as Roya Out® (50% B. subtilis, 2.5% Azadirachta indica, and 2.5% Syzygium aro-maticum), Biogeneser® (Azadirachta indica), and Timorex Gold® (23.8% tea tree oil Melaleuca alternifolia) have been evaluated in field conditions for the control of CLR on sites where the disease has been observed year by year [54,106]. 

Pest control and diseases have two ways of implementation by conservation or intro-duction. One option is by preserving the available biological control agents in nature to obtain ecosystemic benefits, whereas for the introduction method, natural enemies are bred and multiplied in the laboratory or bio-factories [107]. Thus, changing the chemical control methods used in agriculture to sustainable methods controlling phytopathogens should allow maintaining the agroecosystems and ecological biodiversity. Although the applica-tion of chemical synthesis products offers immediate protection, it causes phytopathogen resistance and may bioaccumulate in crops [108]. The cost of using agrochemicals may be reduced up to 60% compared to the use of biological control agents [107].”

Section 7 (Lines 360-375) “7. Future Applications and Challenges

Just as disease appearance may alter the metabolism of coffee plants, causing changes in coffee grain composition and quality of the beverage [109], the different inputs used for disease control may also modify the organoleptic and fruit quality characteristics [110]. In the last years, research on the potential use of essential oils in agriculture has increased considerably. Because of their antimicrobial, fungicidal, and bactericidal properties they are promising candidates for developing new potentially eco-friendly products for an in-tegral pest and disease control. Likewise, their phytotoxic properties may potentiate their application as herbicides in agriculture [111]. Nevertheless, most of the biocontrol appli-cations and studies are limited to greenhouse crops where the environmental conditions are monitored and supervised. In field conditions, extensive studies have also been dedi-cated to evaluate the efficiency of the biological control agents. Nevertheless, this area has been restricted to changes in ecological, genetic, and physiological status of the host and climatological characteristics [108]. In conclusion, alternatives should propose the union of various strategies that protect biocontrol agents from extreme conditions, and their via-bility and bioavailability.”

  1. “The refrences should be cheked since several of them are not related with the content of the sentence, for example in line 39 reference 7.”

ANSWER: References was checked. In line 45 (before line 39) reference 7 was modified according to the content of the manuscript. In line 46,109-121 references were modified.

  1. “Line 296 More recent Reference should be added: https://doi.org/10.1007/s00248-022-02044-2

ANSWER: Reference was added: line 315, reference 98.

  1. “Line 309 this reference is about bacillus lipopeptides not related with ISR.”

ASNWER: The indicated reference was reviewed (BEFORE: line 309, reference 101. NOW: line 329, reference 103.) This article is: Trichoderma spp.: efficient inducers of systemic resistance in plants. In Microbial-mediated induced systemic resistance in plants.

  1. “The english should be checked since there are gramar mistakes such as lines 26, 37, 49, 54, 107 among others.”

ANSWER: The English was checked in lines 32, 43, 55, 60, 122. Other changes were:

  1. The word “of” is removed from the title: “Essential Oils and Antagonistic Microorganisms as Eco-friendly Alternatives for Coffee Leaf Rut Control”.
  2. Line 38 it changed “exist” by “the existence of”.
  3. Line 57 it changed “which” by “they”.
  4. Line 61 the word was added “the”.
  5. Line 80 it changed “affecting” by “affected” and the word was added “This situation”.
  6. Line 81 it changed “it” by “CLR”.
  7. Line 104 it changed “for” by “of”.
  8. Line 105 it changed “is” by “are”.
  9. Line 110 it changed “urediniospores” by “urediniospore”.
  10. Line 112 it changed “period” by “periods”, “of the leaf” by “leaf”.
  11. Line 124 the word was added “the”.
  12. Line 138 the word was added “measures”.
  13. Line139 it changed “kilometers” by “km”.
  14. Line 141 the word was removed “such”.
  15. Line 151 it changed “fungicides” by “fungicide” and the word was added “the”.
  16. Line 152 it changed “being” by “has been”, “application” by “applications”, “the” by “it”, “start” by “starts”.
  17. Line 154 it changed “of the plant” by “plant”.
  18. Line 155 it changed “in” by “on”.
  19. Line 156 the word was added “by”.
  20. Line 158 it changed “of CLR” by “CLR”.
  21. Line 165 the word was added “The”.
  22. Line 171 the word was added “that”.
  23. Line 173 the word was added “of”.
  24. Line 174 the word was added “and”.
  25. Line 207 the word was added “such”.

Round 2

Reviewer 3 Report

The authors have done thecorrections according to the referees comments